# Socioeconomic and ethnic differences in children's vigorous intensity physical activity: a cross-sectional analysis of the UK Millennium Cohort Study

Rebecca Love,[1] Jean Adams,[1] Andrew Atkin,[1,2] Esther van Sluijs[1]

¹Centre for Diet and Activity Research (CEDAR), MRC Epidemiology Unit, University of Cambridge School of Clinical Medicine, Cambridge, UK
²University of East Anglia Faculty of Medicine and Health Sciences, Norwich, UK

**Correspondence to**
Rebecca Love;
rel54@medschl.cam.ac.uk

## ABSTRACT

**Objective** To investigate if daily vigorous physical activity (VPA), adjusted for minutes of moderate physical activity (MPA) performed, differs by socioeconomic position or ethnicity in a large sample of UK children with objectively measured physical activity.

**Design** Nationally representative prospective cohort study.

**Setting** UK children born between 2000 and 2002.

**Participants** 5172 children aged 7–8 with valid accelerometer data for ≥10 hour on ≥3 days, including 1 weekend day.

**Main outcome measures** Time spent in VPA (>3841 counts per min).

**Explanatory measures** Maternal education, annual household Organisation for Economic Co-Operation and Development equivalised income, ethnicity.

**Results** Multivariable linear regression models fitted to explore differences in average daily minutes of VPA (adjusted for MPA, mean accelerometer wear time, season of measurement, age and sex), revealed significantly higher amounts of VPA accumulated as a child's socioeconomic position increased (highest vs lowest level of maternal education: β: 2.96, p: 0.00; annual household equivalised income: β: 0.58, p: 0.00, per £10 000 annual increase). Additionally, children from certain minority ethnicities (Bangladeshi and Pakistani: β: −3.34, p: 0.00; other ethnic groups: β:−2.27, p: 0.02) accrued less daily VPA compared with their white British counterparts.

**Conclusions** The socioeconomic and ethnic patterning of vigorous activity observed in this study mirrors parallel inequalities in rates of childhood obesity. Given the stronger association of VPA with adiposity than of MPA, intensity specific differences may be contributing to widening inequalities in obesity. Accordingly, these findings suggest that the current global focus on overall moderate-to-vigorous intensity activity may mask important behavioural inequalities.

## BACKGROUND

Over the past four decades, global prevalence of childhood obesity has increased 10-fold.[1] Predictions forecast that if current trends continue, by 2030, 60% of the global adult population will be overweight or obese.[2] Given that obesity in childhood is associated

---

**Strengths and limitations of this study**

► This is the first study to investigate socioeconomic and ethnicity-related differences in children's vigorous intensity physical activity behaviour, accounting for moderate physical activity.

► We used data from the Millennium Cohort Study which used a stratified sampling design enabling adequate representation of socioeconomically disadvantaged and ethnic minority children.

► This is the largest available accelerometer dataset of objectively measured physical activity in UK children.

► Boys, certain ethnic minorities (Indian, Pakistani/Bangladeshi and Black Caribbean/African), and children living with only one parent were less likely to provide valid accelerometer data which may have affected the representativeness of our findings.

► The accelerometer measurements utilised are limited in their ability to classify behaviour, detect certain activities (eg, cycling and swimming), upper body movements or changes in terrain.

---

with premature mortality and physical morbidity in adulthood,[3] tackling childhood obesity is increasingly a public health priority for governments.[4]

Of increasing concern are widening inequalities in obesity prevalence. By age 11, UK children from lower income families are three times as likely to be obese than their more advantaged counterparts.[5] These differences progressively worsen with age, fuelling inequality in health across the life course.[6 7] There are also stark ethnic and racial differences in childhood adiposity, with higher rates of obesity within certain ethnic minorities including Black British and Bangladeshi.[8 9] It is suggested that these differences are partially linked to existing disparities in socioeconomic resources between ethnic groups.[9] At present, there is limited understanding of the modifiable factors driving

socioeconomic and ethnicity-related disparities in childhood obesity.

Physical activity is a key behavioural driver of cardiovascular disease risk in children.[10] Evidence of an association with obesity is less clear.[11 12] However, consistent evidence suggests that vigorous physical activity (VPA) is more strongly associated with reduced waist circumference and adiposity relative to lower intensity activity, including moderate physical activity (MPA).[13–17]

Previous research using objectively measured physical activity has demonstrated no socioeconomic patterning in children's adherence to the international guidelines to engage in 'moderate-to-vigorous intensity activity (MVPA) for at least 60 min/day'.[18–20] While there is some evidence of ethnic differences in activity levels,[21] these analyses are confined to the use of aggregated MVPA to quantify activity levels. This focus on MVPA may be why physical activity only explains a small portion of the socioeconomic gradient in overweight risk present within UK children.[22]

The importance of intensity specific differences of MVPA in explaining socioeconomic and ethnic inequalities in health remains underexplored. Considering that moderate physical activity (MPA) and vigorous physical activity (VPA) are accumulated through different types of activities (eg, walking to school vs sport participation),[23] they may be differently distributed in population subgroups. In high-income countries including the UK, children's participation in the organised sports activities that drive VPA have been shown to be socially patterned due to unequal access, support and costs.[24–26] Children from certain ethnic minorities have been shown to face additional barriers to sport participation due to cultural and religious factors, lack of access and parental safety concerns.[27] The need for understanding of these intensity differences is strengthened by evidence suggesting that the age-related decreases in VPA observed annually into adulthood are more pronounced in black and socioeconomically deprived children, than their counterparts.[28] We propose that socioeconomic and ethnic differences in children's higher activity intensity could help to explain well-established socioeconomic and ethnic gradients in obesity. Thus, the objective of this study was to investigate if daily VPA, adjusted for minutes of MPA performed, differs by socioeconomic position or ethnicity in a large sample of UK children aged 7 years.

## METHODS
### Sample
This analysis used data from the Millennium Cohort Study (MCS), a nationally representative, longitudinal study of children born in the UK between September 2000 and January 2002.[29] MCS was developed to track the social, economic and health experiences facing children born at the start of the 21st century. The original cohort included 18 818 children. Data were first collected when cohort participants were 9 months old through a home-based interview (72% of those initially contacted provided information). Subsequent follow-up interviews at ages 3, 5, 7 and 11 years were conducted in the home environment with the main respondent (primarily the natural mother, and where applicable the partner).[30] At age 7 an accelerometer study was conducted with participants. Parental consent and child assent were obtained for participation in the accelerometer study.

To ensure adequate representation of all four UK countries, including disadvantaged and minority populations, a stratified clustered sampling design was used to oversample children living in Wales, Scotland and Northern Ireland, disadvantaged areas, and communities with high proportions of minority ethnic groups. The MCS received ethics approval from the South West and London Multicentre Research Ethics Committees, UK and the Yorkshire Research Ethics Committee. The analyses presented here used data from follow-up 4 at age 7 years (MCS4).

### Patient and public involvement
This analysis is of secondary data collected a decade ago and accordingly the participants included were not involved in the conceptualisation or design of our specific study.

### Physical activity
At MCS4 participating children (n=13 681) were invited to participate in an accelerometer study. Consenting children were sent a preprogrammed Actigraph GT1M accelerometer (Actigraph, Pensacola, Florida, USA), set to record data at 15 s epochs. All data collection and processing was performed in-house, according to predetermined criteria.[31]

Actigraph accelerometers have been shown to be reliable and valid in comparison to measures of physical activity derived from heart rate monitoring, indirect and room calorimetry, and doubly labelled water.[32–34] Children were instructed to wear the accelerometer on an elastic belt around their waist for 7 days throughout all waking hours (except during aquatic activities or contact sports), and to return it by post. Children were additionally asked to complete a monitor wear log to help identify non-wear time. Data were collected over a 15-month period between May 2008 and August 2009.

Data were downloaded using Actigraph V.3.8.3 and processed according to predetermined criteria, using the R statistical package pawacc.[31] Non-wear time, defined as 20 min or more of consecutive zero activity counts, were excluded from the analysis. Counts were separated into sedentary (<100 counts per min (cpm)), moderate (2240–3840 cpm) and vigorous (>3841 cpm) categories.[31] To ensure reliable estimates of activity, the sample was restricted to participants with three valid days of data (≥10 hour/day, including at least one weekend day).[35] The sample was additionally restricted to singleton children. Extreme values above a threshold of ≥11 715 cpm were excluded, as this may indicate a faulty monitor.

## Sociodemographic variables

Considering strong evidence that health and obesity is patterned strongly and independently along both socio-economic[36–38] and ethnic[39 40] lines, our analysis examines the association with both indicators. Information pertaining to socioeconomic position and ethnicity were collected at MCS4. Socioeconomic position was measured using maternal education and equivalised household income. Maternal education captures the socioeconomic circumstances that accrue to a child and is advantageous as it can be applied to mothers irrespective of whether or not they are in paid employment at the time of interview.[41] Maternal education was categorised into five groups: none (qualifications less than those currently expected when leaving school at 16 years); low (qualifications comparable to those currently expected when leaving school at 16 years); medium (qualifications comparable to those currently expected when leaving school at 18 years); high (qualifications greater than medium, but not higher) and higher (any higher educational qualifications).[42] Annual household income was equivalised for household composition based on guidance from the Organisation for Economic Co-Operation and Development (OECD).[29] Ethnicity was parent-reported and categorised in our analyses as: white, any mixed, Indian, Pakistani or Bangladeshi, black or black British, or other. Parents were asked to select from a longer list which ethnic group they identify most with (eg, Black includes those who identified as Black Caribbean, Black African and Black British). The ethnic classifications utilised were based on census categories in accordance with guidelines from the Office for National Statistics.[43]

## Statistical analysis

Multivariable linear regression models were fitted to analyse differences in absolute mean daily minutes of VPA achieved across socioeconomic and ethnic groups, adjusting for mean daily minutes of MPA, mean accelerometer wear time, season of measurement, age and sex. Separate models were run for each exposure variable (maternal education, equivalised household income, ethnicity) to assess the effects of each seperately.[44] Models were also run separately for week and weekend days as there is evidence that children accumulate physical activity differently on weekdays and weekend days.[45] Further, it is possible that different socioeconomic and ethnic subgroups of children engage in different types of activities with distinct weekly patterning (eg, weekend sports). All model residuals were assessed for normality. To investigate effect modification by gender, interactions were run across all models. In sensitivity analyses, additional adjustments for body mass index (BMI) were explored. All analyses were conducted using STATA V.15.1 software, with survey commands used to account for the stratified clustered design of MCS and to obtain robust standard errors.[46 47] Sampling weights adjusted for unit non-response between waves were utilised.

**Table 1** Sociodemographic characteristics of the weighted sample: UK Millennium Cohort Accelerometer Study, fourth follow-up (2008–2009)

| (N=5172) | Level | n (%) |
|---|---|---|
| Gender | Female | 2592 (50.1) |
| | Male | 2580 (49.8) |
| Country | England | 3390 (65.5) |
| | Wales | 676 (13.1) |
| | Scotland | 612 (11.8) |
| | Northern Ireland | 494 (9.6) |
| Weight status | Not overweight | 4296 (83.3) |
| | Overweight | 663 (12.9) |
| | Obese | 196 (3.8) |
| Ethnic group | White | 4543 (87.8) |
| | Mixed | 130 (2.5) |
| | Indian | 115 (2.2) |
| | Pakistani or Bangladeshi | 194 (3.8) |
| | Black or Black British | 127 (2.5) |
| | Other ethnic group | 63 (1.2) |
| Maternal education | No qualifications | 263 (5.1) |
| | Low | 1241 (24.0) |
| | Medium | 787 (15.2) |
| | High | 1965 (38.0) |
| | Higher | 447 (8.6) |
| | Overseas qual. | 108 (2.1) |
| | None of these | 361 (7.0) |
| Equivalised monthly household income (£), mean (SD) | | 1700.9 (943.1) |
| Distribution of valid days | | 6.1 (1.2) |
| Age (years) | | 6.8 (0.39) |

To support the premise of the current analyses that VPA is most strongly associated with adiposity, additional linear regression models were fitted to study differences in BMI-z score by mean daily minutes of VPA and MPA separately, adjusting for accelerometer wear time, age and sex.

## RESULTS

Of the 12 872 children that consented to the accelerometer study, 9772 returned the accelerometers, with a final sample of 6497 children following in-house processing by MCS.[48] Application of our study inclusion criteria resulted in an analytic sample of 5172 children. This drop was predominately driven by our requirement participant's 'three or more valid days', include one weekend day to enable comparisons across weekend and weekdays. On average, children in the weighted sample were 6.8 years of age (SD: 0.4) and 50% female (see table 1). Overall, 14.4% of girls and 11.7% of boys were overweight, while 4.1% and 3.5% were obese. These classifications were

**Table 2** Physical activity characteristics of the weighted sample: UK Millennium Cohort Accelerometer Study, fourth follow-up (2008–2009)

| (N=5172) | All days* | Weekday* | Weekend* |
|---|---|---|---|
| Mean time (min/day) in sedentary behaviour | 397.2 (68.3) | 402.8 (70.8) | 380.5 (91.9) |
| Mean time (min/day) in light activity | 282.7 (40.5) | 282.9 (42.9) | 282.6 (54.5) |
| Mean time (min/day) in moderate activity | 42.5 (13.2) | 42.4 (13.6) | 42.9 (19.1) |
| Mean time (min/day) in vigorous activity | 19.9 (10.6) | 20.1 (11.1) | 19.4 (14.2) |
| Mean time (min/day) worn across all valid days | 742.4 (63.1) | 748.2 (66.1) | 725.4 (88.3) |

*All values are mean (SD).

made through application of the WHO Growth Standards to produce age and gender specific z-scores utilising STATA functions zanthro and zbmicat.[49] The sample included children from each country across the UK (England, Wales, Scotland and Northern Ireland). Sociodemographic and physical activity summary characteristics of the analytic sample are outlined in tables 1 and 2, respectively. These two tables of sample characteristics are based on the weighted sample given that all our analyses are weighted. Dropout analyses showed that those participants included in the analytic sample (n=5172) were more likely to come from a higher income household, have mothers with higher levels of education and be male when compared with participants who provided accelerometer data but did not meet the criteria for the analytic sample (n=1325).

Multivariable linear regression models revealed significant differences in the minutes of daily VPA accumulated across socioeconomic subgroups (table 3, see online supplementary file 1 for full model details). Significantly more minutes of daily VPA was accumulated in each level of maternal education compared with those whose mother indicated 'no qualifications'. This relationship was more pronounced on weekdays than on weekend days. Analyses by equivalised household income supported this, indicating significantly more time spent in VPA with increasing household income. Figure 1 illustrates this effect and shows the proportion of VPA within daily MVPA by categories of activity, stratified into tertiles of equivalised household income. This demonstrates that irrespective of activity level, children from higher affluence families generally accumulate a greater proportion of their daily MVPA from VPA.

Pakistani and Bangladeshi children performed on average over 3 min less daily VPA in comparison to white British children overall, on weekdays and weekend days. This difference was slightly more pronounced on weekdays, versus weekend days. Children from 'other ethnic groups' also accumulated less daily VPA overall and on weekdays (2.2 and 3 min less, respectively). In contrast, children from a mixed ethnic descent accumulated comparatively more minutes of VPA daily across the week and on weekdays, but not weekend days.

There were no significant interactions with gender in any model. Additional adjustments for BMI z-score did

not change the pattern of results (see online supplementary file 2). Supporting multivariable linear regression models for BMI z-score revealed a significant association between daily minutes of VPA and BMI z-score, with a 1 min difference in VPA associated with 0.012 lower BMI z-score (see online supplementary file 3). The association of daily MPA with BMI z-score was statistically significant, but substantially smaller (−0.002).

## DISCUSSION

This study reveals clear socioeconomic and ethnic differences in children's time spent in VPA. Children from less affluent backgrounds, alongside those from certain minority ethnic groups (Bangledeshi and Pakistani, other ethnic groups), accumulated less of their daily activity from vigorous intensity activities. These differences were consistent in both boys and girls, and mirror existing inequalities in childhood adiposity.[50] Although the effect sizes are relatively small, and possibly not clinically relevant at an individual level, we suggest that these differences are relevant at a population level and changes to reduce existing gaps in VPA could have population-level implications for inequalities in adiposity in UK children.[10]

To our knowledge, this is the first study to investigate socioeconomic and ethnicity-related differences in the patterning of children's VPA, accounting for time spent in MPA. While prior studies have investigated activity intensity,[51] these did not account for MPA or have the capacity to investigate subgroups due to sample homogeneity. Accounting for MPA when analysing VPA is critical given differences in the accumulation and distribution of overall activity across individuals. MCS offers the largest available representative accelerometer dataset of UK children from all four countries. The objective measurement of free living activity using accelerometers is critical when comparing subgroups of children,[52] as self-reported data are likely to be differentially biased across these subgroups.[53] An additional strength of the cohort comes from the stratified sampling design which enabled adequate representation of socioeconomically disadvantaged and ethnic minority children. This is further enhanced by the comprehensive sociodemographic information linked through parent interviews. Much previous evidence relies on child-reported assessment of

**Table 3** Multivariable linear regression models for mean minutes of VPA and MPA, respectively, overall for all valid days, on weekdays and on weekend days, by socioeconomic and ethnic subgroups (n=5172) within the Millennium Cohort Accelerometer Study, fourth follow-up (2008–2009)

| Exposure | Level | Overall β-coeff (95% CI) | Weekdays β-coeff (95% CI) | Weekends β-coeff (95% CI) |
|---|---|---|---|---|
| **VPA min/week†** | | | | |
| Ethnic group (ref: white) | Mixed | 1.47* (−0.06 to 3.00) | 1.57* (−0.14 to 3.28) | 0.43 (−1.42 to 2.28) |
| | Indian | 0.74 (−1.19 to 2.67) | 0.58 (−1.23 to 2.39) | 0.76 (−2.18 to 3.70) |
| | Pakistani and Bangladeshi | −3.34*** (−4.66 to −2.03) | −3.45*** (−4.91 to −1.99) | −3.00*** (−4.36 to −1.64) |
| | Black or Black British | −1.67 (−3.96 to 0.61) | −2.08 (−4.47 to 0.32) | −0.03 (−3.44 to 3.37) |
| | Other ethnic group | −2.27** (−4.16 to −0.37) | −3.07*** (−5.31 to −0.83) | −0.14 (−2.28 to 2.00) |
| Maternal education (ref: none) | Low | 1.31** (0.22 to 2.41) | 1.31** (0.11 to 2.52) | 1.34* (−0.06 to 2.73) |
| | Medium | 1.65*** (0.53 to 2.77) | 1.72*** (0.49 to 2.95) | 1.37* (−0.08 to 2.81) |
| | High | 1.81*** (0.77 to 2.86) | 1.88*** (0.73 to 3.03) | 1.63** (0.32 to 2.94) |
| | Higher | 2.96*** (1.45 to 4.46) | 3.04*** (1.39 to 4.70) | 2.80*** (1.06 to 4.53) |
| | Overseas qual. | 2.28* (−0.30 to 4.85) | 2.12 (−0.45 to 4.70) | 2.69 (−0.67 to 6.05) |
| | None of these | −0.45 (−1.77 to 0.87) | −0.60 (−2.02 to 0.83) | −0.07 (−1.77 to 1.64) |
| Equivalised income | Per £ 10 000/year | 0.58*** (0.32 to 0.84) | 0.61*** (0.35 to 0.88) | 0.50*** (0.18 to 0.82) |
| **MPA min/week††** | | | | |
| Ethnic group (ref: white) | Mixed | −2.22** (−4.25 to −0.19) | −2.00* (−4.23 to 0.23) | −2.13 (−5.04 to 0.78) |
| | Indian | −3.94*** (−6.23 to −1.65) | −3.46*** (−5.63 to −1.30) | −4.46** (−7.96 to −0.96) |
| | Pakistani and Bangladeshi | 2.09** (0.33 to 3.85) | 2.00** (0.11 to 3.89) | 2.02* (−0.33 to 4.37) |
| | Black or Black British | 3.01*** (1.07 to 4.95) | 3.10*** (0.81 to 5.39) | 1.77 (−0.96 to 4.49) |
| | Other ethnic group | 1.09 (−2.30 to 4.47) | 1.89 (−1.82 to 5.59) | −1.34 (−5.37 to 2.70) |
| Maternal education (ref:none) | Low | −0.95 (−2.59 to 0.69) | −0.94 (−2.69 to 0.81) | −1.32 (−3.38 to 0.75) |
| | Medium | −1.48* (−3.15 to 0.19) | −1.38 (−3.16 to 0.39) | −1.89* (−4.07 to 0.29) |
| | High | −1.69** (−3.29 to −0.09) | −1.73** (−3.44 to −0.03) | −1.71* (−3.70 to 0.28) |
| | Higher | −2.74*** (−4.55 to −0.92) | −2.71*** (−4.59 to −0.82) | −3.56*** (−5.90 to −1.22) |
| | Overseas qual. | −1.96 (−4.57 to 0.66) | −2.29 (−5.05 to 0.46) | −1.74 (−5.56 to 2.09) |
| | None of these | 1.98* (−0.05 to 4.01) | 1.88* (−0.31 to 4.06) | 2.35* (−0.31 to 5.00) |
| Equivalised income | Per £ 10 000/year | −0.98*** (−1.25 to −0.70) | −0.99*** (−1.27 to −0.71) | −0.92*** (−1.33 to −0.50) |

†All models are adjusted for moderate physical activity, season of measurement, wear time, age and sex.
††All models are adjusted for vigorous physical activity, season of measurement, wear time, age and sex.
*p<0.05; **p<0.01; ***p<0.001.

socioeconomic position, resulting in substantial missing or invalid data, with higher rates of non-response from socioeconomically deprived children.[54]

Like any birth cohort, representativeness of the MCS sample is affected by participant attrition between waves of assessment. Prior analyses of MCS have demonstrated that boys, certain ethnic minorities (Indian, Pakistan/ Bangladeshi and Black Caribbean/African), and children living with only one parent were less likely to provide valid accelerometer data.[55] Dropout analyses showed that the participants included in this analysis were more likely to be male and come from a higher socioeconomic background. It has previously been shown that the presence of a missing-not-at-random mechanism that underestimates

the volume of physical activity during weekend days in this cohort.[56] This mechanism may have influenced our findings. The findings of our models need to be interpreted in consideration of these limitations and our inclusion criteria, specifically our restriction to participants with at least 10 hours of wear time across 3 days. Finally, accelerometers underestimate activity involving vertical movement (eg, cycling) and those for which the accelerometer was not to be worn (eg, aquatic activities or contact sports). If these behaviours are also socioeconomically or ethnically patterned, this may have led to an under or overestimation of the true associations.

Our supplementary analyses support the well-established notion that VPA is more strongly associated with

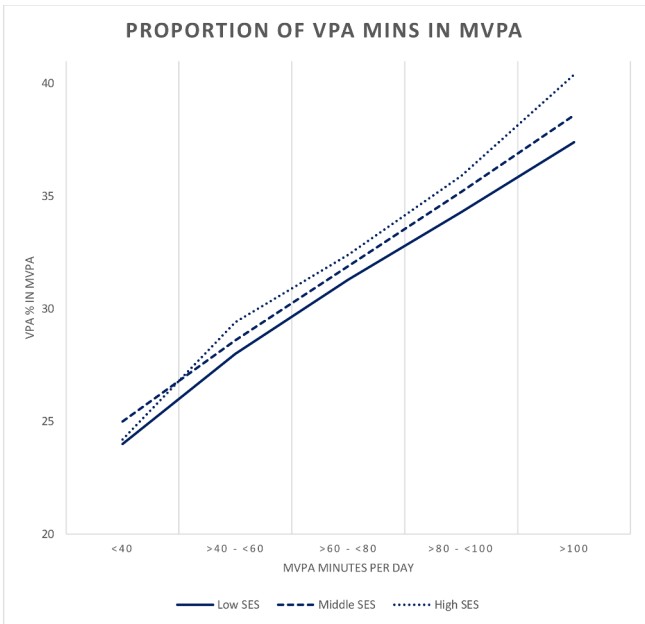

**PROPORTION OF VPA MINS IN MVPA**

**Figure 1** Proportion of VPA in daily MVPA (min), by level of activity with participants grouped by low, middle and high socioeconomic status. Tertiles of household equivalised income is used as indicator of socioeconomic status.

BMI z-score than MPA,[13 15–17] and point to a relevant effect size at a population level.[10] There are multiple reasons why the differences in VPA observed may exist. Due to unequal access and costs, the organised contexts through which children accumulate VPA have been shown to result in differences in participation between more and less advantaged subgroups of children.[24–26] These findings are however based on questionnaire-based assessments which capture sports and organised activity more accurately than other types of activity.[57] Additional factors, including parental perceptions of time commitments, and the limited variety of activities accessible, are significant factors linked to low levels of vigorous physical activity in low income families.[58] Furthermore, differences in home and family support for physical activity have been demonstrated between ethnic groups.[27] For instance, the presence of cultural and religious barriers have been found to impede participation in organised activity among Pakistani and Bangladeshi children, who in this analysis had the lowest levels of VPA. Our findings also demonstrate that differences in VPA between children from high and low maternal education families are more pronounced during the week. This may result from factors such as longer, inconsistent work hours within low-income jobs. These differences and differential barriers should be considered when developing interventions to increase VPA, and potentially impact obesity prevalence, in all groups. Furthermore, this finding highlights the potential of school-based and after-school sports clubs to contribute to a solution for reducing existing behavioural inequalities. Despite known differences in boys' and girls' MVPA,[59 60] also in this cohort,[21] the lack of gender interaction in this analysis reveal that both girls'

and boys' participation in VPA are equally affected by socioeconomic and ethnic factors.

International physical activity guidelines for children focus on MVPA in part because increases in MVPA are hypothesised to be easier to achieve than VPA at a population level.[61] Our findings indicate that by utilising the aggregate measure of MVPA we may be overlooking significant differences in the relative participation in MPA and VPA between population subgroups and tolerating substantial inequalities in the most important segment of physical activity for health outcomes. UK activity guidelines additionally recommend that children 'minimise the amount of time spent being sedentary for extended periods' and that 'vigorous intensity activities be incorporated at least three times a week'. Our results provide further empirical evidence to support the findings of Richards et al. (2015),[62] and their accompanying call to place more attention on the VPA component of guidelines to ensure health benefits. Further evidence suggests that childhood participation in sporting activities is vital to the development of fundamental motor skills, which strongly predict physical activity and weight status both in childhood and throughout adulthood.[63] To lay a foundation for lifelong activity participation it is critical that children, especially those from disadvantaged backgrounds, are provided with sufficient opportunities to develop fundamental motor skills. However, there is currently insufficient evidence concerning the most appropriate daily dose of VPA, and how to effectively promote VPA across population subgroups. Research efforts are needed to develop effective interventions for increasing VPA.

## CONCLUSION

We found that the amount of vigorous intensity activity accumulated was socioeconomically and ethnically patterned in 7-year-old UK children, mirroring known inequalities in adiposity.[50] These findings suggest that the current central focus of physical activity guidelines, and accordingly interventions, on the promotion of aggregate measures of MVPA, may be masking behavioural differences that may have an influential role in widening inequalities in obesity between more and less advantaged subgroups. In efforts to combat rising and widening childhood obesity rates, our results suggest a need for a greater focus on the promotion of VPA in health promotion efforts, particularly for those children from more disadvantaged backgrounds.

**Acknowledgements** The co-operation of the participating families is gratefully acknowledged. The fourth sweep of the MCS was funded by grants to Professor Heather Joshi, former director of the study, from the Economic and Social Research Council and a consortium of government funders. The current director is Professor Lucinda Platt. The authors acknowledge the Centre for Longitudinal Studies, Institute of Education, for the use of these data; the UK Data Service for making them available; and the MRC Centre of Epidemiology for Child Health (grant reference G0400546), Institute of Child Health, University College London, for creating the accelerometer data resource, which was funded by the Wellcome Trust (grant reference 084686/ Z/08/A). The institutions and funders acknowledged bear no responsibility for the analysis or interpretation of these data. We also would like

to thank Stephen Sharp at the MRC Epidemiology Unit for his statistical insights and input on the models.

**Contributors** RL and EvS designed the study. JA and AA provided inputs on preliminary results and contributed to redeveloping the statistical models. RL conducted all of the analyses and drafted the manuscript. All authors contributed to the interpretation of the results and critically reviewed the manuscript. All authors read and approved the final manuscript. RL is the guarantor and responsible for the overall content.

**Funding** Funding for this study and the work of all authors was supported, wholly or in part, by the Centre for Diet and Activity Research (CEDAR), a UKCRC Public Health Research Centre of Excellence (087636/Z/08/Z; ES/G007462/1; MR/K023187/1). Funding from the British Heart Foundation, Department of Health, Economic and Social Research Council, Medical Research Council, and the Wellcome Trust, under the auspices of the UK Clinical Research Collaboration, is gratefully acknowledged. Rebecca Love is funded by a Gates Cambridge Scholarship. The work of Esther van Sluijs was supported by the Medical Research Council (MC_UU_12015/7). No funders were involved in the production of this manuscript.

**Competing interests** None declared.

**Provenance and peer review** Not commissioned; externally peer reviewed.

**Data sharing statement** This is a secondary data analysis based on the Millennium Cohort Study data. The data and access policy are available online here: https://beta.ukdataservice.ac.uk/datacatalogue/series/series?id=2000031#!/access.

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
