## [Reviewer comments · BMJ Open]

ARTICLE DETAILS

TITLE (PROVISIONAL)	Socio-economic and ethnic differences in children's vigorous intensity physical activity: a cross-sectional analysis of the UK Millennium Cohort Study
AUTHORS	Love, Rebecca; Adams, J; Atkin, Andrew; van Sluijs, Esther

VERSION 1 - REVIEW

REVIEWER	Marco Geraci University of South Carolina, USA
REVIEW RETURNED	01-Dec-2018

GENERAL COMMENTS	The authors investigated socio-economic and ethnic patterning of vigorous physical activity (VPA). This is sensible given that VPA is more strongly associated with adiposity than MVPA. Comments - p.7 l.54 "To ensure reliable estimates of activity, the sample was restricted to singleton children with 3 valid days of data". The restriction to singletons is not relevant to the premise (reliability). The exclusion of multiples (twins and triplets) must be stated in a separate sentence.- p.7 l.57. "children with ≥ 150 mean mins of MVPA per day". It seems there is a typo. Perhaps the authors mean "< 150". But in either case, explain why this exclusion.- p.8 l.30. Clarify what the outcome is. Total VPA, mean VPA?- p.8 l.47. Did the authors use MCS sampling weights adjusted for unit nonresponse between waves?- p.8 l.49. Please add a reference for "robust standard errors".- It is profoundly wrong to adjust for MVPA since VPA is the outcome.- The authors adjusted the models for wear time. Did they also constrain the model to have a zero intercept? If wear time is zero, then VPA must be zero. Is adjusting models for wear time a good approach? Or would standardising the outcome a better approach?
--

- Why exposure variables (education, income, ethnicity) were included in separate models and not in the same model?

- p.8 l.42 "VPA was non-normally distributed and therefore all regression models were additionally run using a log-transformed VPA outcome variable". It doesn't matter whether VPA is normally distributed or not. The authors must check the model's residuals. Are they approximately normal with constant variance?

- p.8 l.44 "Results were essentially unchanged thus non-transformed analyses are presented here for ease of interpretation". What do authors mean by "essentially unchanged"? The parameters of a log-transformed model are different so results cannot be the same. The main point here is to check the residuals. If these do not satisfy the linear model assumptions, then consider a median regression model which does not depend on distributional assumptions about the error term (e.g., see Geraci, *Statistical Methods in Medical Research*, 25(4), 1393-1421; Griffith et al, *Longitudinal and Life Course Studies*, 7(2), 124-14).

- Table 3. Please report the intercept as well as the coefficients of all the other variables in the model (moderate physical activity, season of measurement, wear time, age and sex).

- Where appropriate, show the distribution of valid days (proportion of children with X number of valid days).

- "Drop-out analyses showed that the participants included in this analysis were more likely to be male and come from a higher socioeconomic background. Although this may affect the representativeness of our findings, this is unlikely to have affected the associations observed." It seems the authors are implying a missing at random mechanism. I suggest they read the paper by Geraci and Farcomeni (*Journal of the Royal Statistical Society C*, 65(1), 51-75) who found evidence of a missing-not-at-random mechanism (nonignorable) using the same data. In particular, they found that "During weekdays, the predicted probability that data are missing was lower with higher volumes of activity as measured by total counts. Intuitively, this could be explained by the lower occurrence of non-wear periods when more activity is recorded. The negative coefficients for sedentary time on the one hand, but positive for steps and MVPA on the other, are perhaps a consequence of higher compliance rates observed between Monday and Friday, hence when children might be less active because they are involved in day-to-day routines. In contrast, opposite associations were observed during weekend days, when the fraction of missing values tend to be much higher. These results could be interpreted as a consequence of a process by which children are more likely not to follow the study protocol if they do not participate in moderate-to-vigorous activities. This clearly leads to underestimating the volume of physical activity and the proportion of sedentary time, but only during weekend days. This finding might also explain the association that was found by Rich et al. (2013, *BMJ Open*) between lack of exercise and non-response in the MCS data."

- I find the table in Supplementary file A a bit confusing. I counted 9 different models. Is that correct? What is the purpose of having all those models?

REVIEWER	Rebecca Hasson University of Michigan
REVIEW RETURNED	22-Dec-2018

GENERAL COMMENTS	The manuscript entitled: socio-economic and ethnic differences in children's vigorous intensity physical activity: a cross-sectional analysis of the UK Millennium Cohort Study examines differences in patterning of children's activity using objective accelerometer measurements. They determined that socioeconomic and ethnic patterning of vigorous activity are present and mirror inequalities in rates of childhood obesity. While the findings are of interest to the readers of BMJ, the lack of conceptualization of how socioeconomic position and ethnicity differentially impact children's physical activity behaviors along with the absent information about potential differences in sedentary behavior, LPA, and MPA dampens my enthusiasm for the publication of this paper at the this time. My specific comments are below: ABSTRACT: The abstract states that the objective of this paper is to examine differences in intensity patterning but only information on VPA is provided. It isn't until the conclusion of the abstract that a rationale is provided for why only VPA was assessed. For the abstract and the entire paper it would be helpful to examine intensity patterning across the spectrum of activity levels to gain a clear understanding of differences in physical activity by SES and ethnicity. INTRODUCTION: The authors state in the first paragraph that childhood obesity is a major global problem but then state in the fourth paragraph that MVPA only explains a small portion of the SES gradient in overweight children. The authors then state that VPA may be a stronger predictor of obesity in children. Rather than speculate about these relationships the authors should actually examine these relationships in their analysis (1 examine SES and ethnic differences in SED, LPA, MPA, and VPA; 2 determine if these differences in physical activity are predicting differences in obesity prevalence in their current sample of children). The authors need to expand the discussion about how physically activity is accumulated throughout the day may differ by SES and ethnicity. Are low-SES children more likely to walk to school thereby accumulating more MPA rather than play sports (i.e. VPA)? If so, this will drive home the point that examining how physical activity is accumulated may play an important role in shaping disparities in health outcomes. The authors need to provide a clear rationale for why they expect SES to have a differential impact on physical activity patterning compared to race/ethnicity. Why do the readers expect to see differences in physical activity by ethnicity, independent of SES? See Bravemen et al 2010 "Socioeconomic disparities in health in the United States: what the patterns tell us". While this paper discusses these social constructs in the context of the US, it may help the authors better articulate why its important to look at both SES and ethnicity and how these findings may shape future policies and interventions.
--

As a follow-up to the previous comment, authors also need to explain how they are conceptualizing education and income and how these two variables may differentially impact physical activity behaviors in youth. How or why would maternal education differentially impact VPA compared to household income.

In the last sentence of the Introduction the authors state that they are adjusting for MPA. Again without a clear conceptualization of how the authors think physical activity patterns differ, it is unclear why authors are adjusting for MPA.

METHODS: The authors state that only the 7 year old cohort is examined in this analysis but also mention that data was collected for children ages 3, 5, 7, and 11. Was accelerometer data only collected during the 7-year old data collection period? If so, authors should state this. If not, it would also seem appropriate to complete a longitudinal analysis of changes in pa patterning from ages 7-11 as this is a critical period with physical activity declines dramatically.

Authors should provide a rationale for using 15-second epoch.

Authors should also provide a rationale for excluding very high active children. How many of the children in the sample reported this high level of activity. Nader et al 2008 "MVPA from ages 9 to 15 years" demonstrated that 9 year old children averaged 182 minutes of MVPA per weekday.

Were activity logs taken for times when children were not wearing their accelerometers?

For international readers, authors may want to define the ethnic categories of "black" and "black British".

In the statistical analysis section authors state that they adjust for BMI. Because they do in fact have this data, I suggest that they also look at whether differences in VPA predict BMI in their cohort of children.

RESULTS: The authors provide information on differences between the children with and without complete accelerometer data. It would also be useful to include comparisons for children included in the analysis and children without any accelerometer data at all.

Line 45 of the results, please state how many minutes less daily VPA that black and black British children were accumulating.

DISCUSSION: A primary issue with the current analysis is that the authors fail to provide a compelling argument for the clinical significance of 3 fewer minutes of VPA. Three minutes in and of itself is not substantial but if you can demonstrate these three minutes are in fact associated with increased BMI in your cohort it would provide clinical significance to these findings.

The authors should provide some discussion regarding the lack of gender differences in the present cohort as this is contrary to most current research.

	The authors note that the UK activity guidelines recommend that children "minimise the amount of time spent being sedentary . . . " This is another missed opportunity for this paper to explore as sedentary time may in fact be more predictive of obesity given the increased feeding that using occurs with these behaviors.
--	---

VERSION 1 – AUTHOR RESPONSE

Reviewer 1	Response
- p.7 l.54 "To ensure reliable estimates of activity, the sample was restricted to singleton children with 3 valid days of data". The restriction to singletons is not relevant to the premise (reliability). The exclusion of multiples (twins and triplets) must be stated in a separate sentence.	Thank you for the suggestion. This change has been made and is outlined below: Line 127-130: "To ensure reliable estimates of activity, the sample was restricted to participants with 3 valid days of data (≥ 10 h/day, including at least 1 weekend day).(Mattocks et al., 2008) The sample was additionally restricted to singleton children."
- p.7 l.57. "children with ≥ 150 mean mins of MVPA per day". It seems there is a typo. Perhaps the authors mean "< 150". But in either case, explain why this exclusion.	On reflection we removed this exclusion to restrict highly active children. The maximum minutes of MVPA within those meeting our valid days and wear time restriction was an average of 263.1 minutes/day which is not an infeasible amount of MVPA. We have therefore re-run all analyses and updated Tables 1-3.
- p.8 l.30. Clarify what the outcome is. Total VPA, mean VPA?	The outcome is mean daily minutes of VPA across all valid days. We have worked to clarify this in multiple places: Line 153-157: "Multivariable linear regression models were fitted to analyse differences in absolute mean daily minutes of VPA achieved across socio-economic and ethnic groups, adjusting for mean daily minutes of MPA, mean accelerometer wear time, season of measurement, age and sex." Line 188-191 "Multivariable linear regression models revealed significant differences in the mean minutes of daily VPA accumulated across socio-economic subgroups (Table 3). Significantly more minutes of daily VPA was accumulated in each level of maternal education compared to those whose mother indicated 'no qualifications'."

- p.8 l.47. Did the authors use MCS sampling weights adjusted for unit nonresponse between waves?	Yes, the models have all been adjusted using the sampling weight adjusted for non-response up until the 4th wave data utilized in this analysis. This has been clarified at Line 164-165 “Sampling weight adjusted for unit nonresponse between waves were utilized.”
- p.8 l.49. Please add a reference for "robust standard errors".	References have been added as outlined here: Line 163-164: “All analyses were conducted using STATA 15.1 software, with survey commands used to account for the stratified clustered design of MCS and to obtain robust standard errors.(Huber, 1967; StataCorp, 2010)”
- It is profoundly wrong to adjust for MVPA since VPA is the outcome.	We agree that it is inappropriate to adjust for MVPA. We have removed the reference to the analyses adjusting for MVPA.
- The authors adjusted the models for wear time. Did they also constrain the model to have a zero intercept? If wear time is zero, then VPA must be zero. Is adjusting models for wear time a good approach? Or would standardising the outcome a better approach?	We did not constrain our models to have a zero intercept. Our inclusion criteria restricts participants to those with 10 hours of wear time per day, over three days. Resulting from this the mean wear time of included participants is 733.5 minutes/day (SD: 63.1) with the lowest observed wear time being 615.2 minutes/day and the highest 1177.9 minutes/day. We agree that extrapolating the findings from this model to individuals with wear time outside of this range would be unreasonable and have limited the conclusions we have drawn to this. Line 280-282: “The findings of our models need to be interpreted in consideration of these limitations and our participant inclusion criteria, specifically our restriction to participants with at least 10 hours of wear time across three days.”
- Why exposure variables (education, income, ethnicity) were included in separate models and not in the same model?	Exposure variables were included in separate models as we were interested in the effects of each one in isolation. We have highlighted this decision in the methods sections. Line 155-157: “Separate models were run for each exposure variable (maternal education, equalised

	household income, ethnicity) to assess the effects of each one in isolation.(Green and Popham, 2019)”
- p.8 l.42 "VPA was non-normally distributed and therefore all regression models were additionally run using a log-transformed VPA outcome variable". It doesn't matter whether VPA is normally distributed or not. The authors must check the model's residuals. Are they approximately normal with constant variance?	Thank you for raising this issue. We agree that the results were indeed different, although the conclusions drawn from the models with similar. We did check whether the models with non-transformed VPA met the criteria for linear regression. The residuals were normally distributed. This is outlined in the methods on Line 160. The log-transformed models have been removed from the appendix and the reference to them in the text.
- p.8 l.44 "Results were essentially unchanged thus non-transformed analyses are presented here for ease of interpretation". What do authors mean by "essentially unchanged"? The parameters of a log-transformed model are different so results cannot be the same. The main point here is to check the residuals. If these do not satisfy the linear model assumptions, then consider a median regression model which does not depend on distributional assumptions about the error term (e.g., see Geraci, Statistical Methods in Medical Research, 25(4), 1393-1421; Griffith et al, Longitudinal and Life Course Studies, 7(2), 124-14).	
Table 3. Please report the intercept as well as the coefficients of all the other variables in the model (moderate physical activity, season of measurement, wear time, age and sex).	We now provide the full details of each model (Overall, Weekdays and Weekends for both daily VPA and MPA) in the appendix with coefficients for all variables included in models, the intercept and R ² . This is noted in the results at line 188-190: “Multivariable linear regression models revealed significant differences in the minutes of daily VPA accumulated across socio-economic subgroups (Table 3, see supplementary file 1 for full model details).”
- Where appropriate, show the distribution of valid days (proportion of children with X number of valid days).	The distribution of valid days is now included in Table 1.
- "Drop-out analyses showed that the participants included in this analysis were more likely to be male and come from a higher socioeconomic background. Although this may affect the representativeness of our	Thank you for highlighting these findings that we were not aware of. We have incorporated these findings in our limitations section. Line 276-282:

findings, this is unlikely to have affected the associations observed."

It seems the authors are implying a missing at random mechanism. I suggest they read the paper by Geraci and Farcomeni (Journal of the Royal Statistical Society C, 65(1), 51-75) who found evidence of a missing-not-at-random mechanism (nonignorable) using the same data. In particular, they found that "During weekdays, the predicted probability that data are missing was lower with higher volumes of activity as measured by total counts. Intuitively, this could be explained by the lower occurrence of non-wear periods when more activity is recorded. The negative coefficients for sedentary time on the one hand, but positive for steps and MVPA on the other, are perhaps a consequence of higher compliance rates observed between Monday and Friday, hence when children might be less active because they are involved in day-to-day routines. In contrast, opposite associations were observed during weekend days, when the fraction of missing values tend to be much higher. These results could be interpreted as a consequence of a process by which children are more likely not to follow the study protocol if they do not participate in moderate-to-vigorous activities. This clearly leads to underestimating the volume of physical activity and the proportion of sedentary time, but only during weekend days.

This finding might also explain the association that was found by Rich et al. (2013, BMJ Open) between lack of exercise and non-response in the MCS data."

"Drop-out analyses showed that the participants included in this analysis were more likely to be male and come from a higher socioeconomic background. A prior analysis in this same MCS dataset found the presence of a missing-not-at-random mechanism that underestimates the volume of physical activity during weekend days (Geraci and Farcomeni, 2016). This mechanism may have influenced our findings including the stronger differences. The findings of our models need to be interpreted in consideration of these limitations and our inclusion criteria, specifically our restriction to participants with at least 10 hours of wear time across 3 days."

- I find the table in Supplementary file A a bit confusing. I counted 9 different models. Is that correct? What is the purpose of having all those models?

Yes, we previously had included multiple models demonstrating differences in variable adjustments. We have removed these and in their place is:

- 1: Full model details with B-coefficients and 95% confidence intervals for all variables included within main models, also including intercept and R2.
- 2: Multivariable linear regression models for BMI z-score by daily minutes of VPA and MPA. [Note: these have been added for this revision based on feedback from our second reviewer below].
- 3: Full models showing additional adjustments for BMI.

Reviewer 2	
ABSTRACT: The abstract states that the objective of this paper is to examine differences in intensity patterning but only information on VPA is provided. It isn't until the conclusion of the abstract that a rationale is provided for why only VPA was assessed. For the abstract and the entire paper it would be helpful to examine intensity patterning across the spectrum of activity levels to gain a clear understanding of differences in physical activity by SES and ethnicity.	Our central question and focus within this paper is on ethnic and SES differences in the accumulation of higher intensity activity (VPA) within the context of the other activity undertaken (MPA). This is because we are interested specifically in the reduced adiposity and greater health benefits linked to higher intensity activity in light of existing and widening obesity inequalities in children. The abstract guidelines of BMJ Open do not allow for background information to be included, but we have revised the objective in the abstract to highlight our focus on VPA. Line 3-4: “Objective: “To investigate if daily vigorous physical activity, adjusted for minutes of moderate physical activity performed, differs by socio-economic position or ethnicity in a large sample of UK children with objectively measured physical activity.” We would like to retain the focus of the paper on VPA and have concerns that including outcome variables across the intensity spectrum would make the paper unfocussed. However, based on your comments below we have now included complementary models for MPA as the dependent variable in the main results table presented on page 15.
INTRODUCTION: The authors state in the first paragraph that childhood obesity is a major global problem but then state in the fourth paragraph that MVPA only explains a small portion of the SES gradient in overweight children. The authors then state that VPA may be a stronger predictor of obesity in children. Rather than speculate about these relationships the authors should actually examine these relationships in their analysis (1 examine SES and ethnic differences in SED, LPA, MPA, and VPA; 2 determine if these differences in physical activity are predicting differences in obesity prevalence in their current sample of children).	Thank you for the suggestion. We agree that these are important questions, but we think that these go beyond the focus of the current paper as they address different research questions. We also felt that there is strong existing evidence linking VPA differences to adiposity. We do however agree with the point you raise that we have all the data to be able to show this within our sample. We have now conducted an analysis to study differences in BMI-zscore by mean daily minutes of VPA and MPA separately, using linear regression models adjusted for accelerometer wear time, age and sex. We have discussed these in the results and discussion (see below) and have included the chart in the Appendix (File 3).

	We have also made changes to better articulate that robust evidence exists that VPA is a better predictor of obesity in children than lower intensity activities: Introduction line 63-65: “However, consistent evidence suggests that vigorous physical activity (VPA) is more strongly associated with reduced waist circumference and adiposity relative to lower intensity activity, including moderate intensity activity (MPA).(Gutin et al., 2005; Steele et al., 2010; Carson et al., 2013; Pavey et al., 2013; Owens, Galloway and Gutin, 2017)” Methods line 166-168: “To support the premise of the current analyses that VPA is most strongly associated with adiposity, additional linear regression models were fitted to study differences in BMI z-score by mean daily minutes of VPA and MPA separately, adjusting for accelerometer wear time, age and sex.” Results line 205-210: “Supporting multivariable linear regression models for BMI z-score revealed a significant association between daily minutes of VPA and BMI z-score, with a 1-minute difference in VPA associated with 0.012 lower BMI z-score (See Supplementary File C). The association of daily MPA with BMI z-score was statistically significant, but substantially smaller (-0.002).” Discussion, lines 388-290: “Our supplementary analyses support the well-established notion that VPA is more strongly associated with BMI z-score than MPA,(Gutin et al., 2005; Steele et al., 2010; Carson et al., 2013; Owens, Galloway and Gutin, 2017) and point to a relevant effect size at a population level (Elelund et al. 2005).”
The authors need to expand the discussion about how physically activity is accumulated throughout the day may differ by SES and ethnicity. Are low-SES children more likely to walk to school thereby accumulating more MPA rather than play sports (i.e. VPA)? If so, this will drive home the point that examining	We thank the reviewer for this suggestion and have changed a paragraph in our introduction to better articulate this Line 73 - 80: “The importance of intensity specific differences of MVPA in explaining socioeconomic and ethnic

how physical activity is accumulated may play an important role in shaping disparities in health outcomes.	inequalities in health remains underexplored. Considering that moderate physical activity (MPA) and vigorous physical activity (VPA) are accumulated through different types of activities (e.g. walking to school vs. sport participation),(Butte et al., 2017) they may be differently distributed in population subgroups. In high-income countries including the UK, children’s participation in the organized activities that drive VPA have been shown to be socially patterned due to unequal access, support and costs.(Hardy et al., 2010; Fernandes et al., 2012; Marques, Ekelund and Sardinha, 2015) Children from certain ethnic minorities have been shown to face additional barriers to sport participation due to cultural and religious factors, lack of access and parental safety concerns.(Trigwell et al., 2015) “
The authors need to provide a clear rationale for why they expect SES to have a differential impact on physical activity patterning compared to race/ethnicity. Why do the readers expect to see differences in physical activity by ethnicity, independent of SES? See Bravemen et al 2010 "Socioeconomic disparities in health in the United States: what the patterns tell us". While this paper discusses these social constructs in the context of the US, it may help the authors better articulate why its important to look at both SES and ethnicity and how these findings may shape future policies and interventions.	While ethnicity and socioeconomic status often overlap substantially, we are interested in differences in intensity patterning within each separately. We have not included indicators of socioeconomic status and ethnicity within the same models so we can address the question of differences in physical activity by ethnicity separately from socioeconomic resources. Given clear differences in overweight and obesity for example within certain ethnic subgroups we were interested if we see parallel differences in the patterning of activity intensity irrespective of and without adjusting for the income of participants. We agree and have worked to articulate why it is important to look both at socioeconomic status and ethnicity in today’s context. Methods Line 133-135: “Considering strong evidence that health and obesity is patterned strongly and independently along both socioeconomic (Zhang and Wang, 2004; Shrewsbury and Wardle, 2008; Braveman et al., 2010) and ethnic (Harding et al., 2008; Cole, Law and MCS Child Health Group, 2009) lines, our analysis examines the influence of both indicators.”
As a follow-up to the previous comment, authors also need to explain how they are conceptualizing education and income and how these two variables may differentially impact physical activity behaviors in youth.	Maternal education and household income are two measures commonly used as indicators of socioeconomic status in child populations. We choose to use both to add strength to our analysis and conclusion on if physical activity patterning

How or why would maternal education differentially impact VPA compared to household income.	differs by socioeconomic status considering both of their frequent use across the academic literature. Maternal education and household income measure different aspects of SES and are not necessarily strongly related. Within our analysis we are not interested in the comparative effects. For this reason, we have not hypothesized or discussed why differences may exist between the two. This choice is addressed within our methods section.
In the last sentence of the Introduction the authors state that they are adjusting for MPA. Again without a clear conceptualization of how the authors think physical activity patterns differ, it is unclear why authors are adjusting for MPA.	As stated above, we have revised our introduction to more clearly conceptualize why we hypothesize that activity patterns differ between subgroups. We adjusted for MPA to be able to analyze and draw conclusions on differences in VPA accumulation in the context of overall MVPA. Without adjusting for MPA we would be unable to draw conclusions on differences in the proportion of VPA accumulated in the context of their overall activity behavior. To better communicate our rationale, analytical approach and findings we now include Figure 1. This shows the proportion of VPA within overall MVPA by categories of activity (ranging from those with avg. daily minutes of MVPA less than 40 to greater than 100). The figure demonstrates that children from high socioeconomic backgrounds consistently accumulate proportionally higher amounts of VPA than children from low socioeconomic backgrounds irrespective of the level of MVPA. We have included this figure in the results and at Line 194-197 to better support our analysis/findings. “Figure 1 illustrates this effect and shows the proportion of VPA within daily MVPA by categories of activity, stratified into tertiles of equivalised household income. This demonstrates that irrespective of activity level, children from higher affluence families generally accumulate a greater proportion of their daily MVPA from VPA.”
METHODS: The authors state that only the 7 year old cohort is examined in this analysis but also mention that data was collected for children ages 3, 5, 7, and 11. Was accelerometer data only collected during the 7-year old data collection period? If so, authors should state this. If not, it would also seem appropriate to complete a longitudinal analysis of changes in pa patterning from ages 7-11 as this is a critical period with	Within the MCS unfortunately accelerometer data was only collected during the fourth follow at age 7 so we are unable to look at longitudinal changes. This is communicated to readers at Line 98-100: Line 105-109 in the Methods “The analyses presented here used data from follow-up 4 at age 7 years (MCS4) when an

physical activity declines dramatically.	accelerometer study was conducted with participants.”
Authors should provide a rationale for using 15-second epoch.	In the MCS accelerometer processing was done by the MCS study team and accordingly we are restricted to the in-house processing decisions (Marco Geraci et al., 2012). The 15 second sampling epoch chosen aligns with current research demonstrating its accuracy within a child population (Aibar and Chanal, 2015; Banda et al., 2016).
Authors should also provide a rationale for excluding very high active children. How many of the children in the sample reported this high level of activity. Nader et al 2008 "MVPA from ages 9 to15 years" demonstrated that 9 year old children averaged 182 minutes of MVPA per weekday.	As mentioned in our response to Reviewer 1 in relation to a similar comment, on reflection we reconsidered and removed this exclusion to restrict highly active children. The maximum number of minutes of MVPA within those meeting our valid days and wear time restriction was an average of 263.1 minutes/day which we agree is a reasonable amount. Given this we have gone back and re-run the characteristics and models updating Tables 1-3.
Were activity logs taken for times when children were not wearing their accelerometers?	Yes children/parents were asked to record when they put on and took off the monitor as well as time spent engaged cycling and swimming. However, this information was not used to adjust the accelerometer outcomes for explained non-wear periods as this is likely to introduce bias and error. We have added this detail into our methods: Line 119-122: “Children were instructed to wear the accelerometer on an elastic belt around their waist for 7 days throughout all waking hours (except during aquatic activities or contact sports), and to return it by post. Children were asked to complete a monitor wear log. Data was collected over a 15-month period between May 2008 – August 2009.”
For international readers, authors may want to define the ethnic categories of "black" and "black British".	Thank you for highlighting that this may cause confusion. The categories used were based on census categories and included those identifying as Black British, Black Caribbean or Black African. Line 145-150: “Ethnicity was parent-reported and categorised in our analyses as: white, any mixed, Indian, Pakistani or Bangladeshi, black or black British, or other. Parents were asked to select from a longer list

	which ethnic group they identify most with (E.g. Black includes those who identified as Black Caribbean, Black African and Black British). The ethnic classifications utilized were based on census categories in accordance with guidelines from the Office for National Statistics.(Office for National Statistics, 2003)”
In the statistical analysis section authors state that they adjust for BMI. Because they do in fact have this data, I suggest that they also look at whether differences in VPA predict BMI in their cohort of children.	Thank you for this suggestion. As highlighted earlier, as supporting analyses we ran additional models investigating if differences in VPA were associated with differences in BMI z-score in the cohort. These findings are discussed in the results and discussion (as outlined above) and presented as Supplementary File 3.
Line 45 of the results, please state how many minutes less daily VPA that black and black British children were accumulating.	With the changes outlined above to inclusion and our subsequent analyses we re-ran all of our models. This very slightly changed our findings so that the difference in VPA amongst black and black British children was no longer significant. However, children from ‘Other ethnicities’ performed significantly less overall VPA and on weekdays. We have highlighted this change, with the minutes specified in the results: At Line 198-203: “Pakistani & Bangladeshi children performed on average over 3 minutes less daily VPA in comparison to white British children overall, on weekdays and weekend days. This difference was slightly more pronounced on weekdays, versus weekend days. Children from ‘other ethnic groups’ also accumulated less daily VPA overall and on weekdays (2.2 and 3 minutes less, respectively). In contrast, children from a mixed ethnic descent accumulated comparatively more minutes of VPA daily across the week and on weekdays, but not weekend days.”
DISCUSSION: A primary issue with the current analysis is that the authors fail to provide a compelling argument for the clinical significance of 3 fewer minutes of VPA. Three minutes in and of itself is not substantial but if you can demonstrate these these three minutes are in fact associated with increased BMI in your cohort it would provide clinical significance to these findings.	The change and relevance we are interested in are at the population level. Although the effect size at an individual level is relatively small, as we discuss in our discussion a shift in the population distribution of vigorous intensity activity, even by a small margin, would have benefits on a population level. As discussed above we ran the additional models suggested to be able to demonstrate the link between vigorous activity and adiposity within this specific sample. This size of this association points

	to a relevant effect size at a population-level. We have re-iterated this within our discussion.
The authors should provide some discussion regarding the lack of gender differences in the present cohort as this is contrary to most current research.	We agree that there are clear gender differences in activity participation, which have also been demonstrated within this cohort (Griffiths). However, the purpose of our gender interaction was to investigate whether the association between SES indications/ethnicity and VPA accumulation differed by gender, not to explore gender differences per se. The lack of significant interactions demonstrate that girls' and boys' VPA participation is similarly affected by socioeconomic status and ethnicity, which is a novel finding. Line 306-308: “Despite known differences in boys' and girls' MVPA,(Troost et al., 2002; Hallal et al., 2012) and in this cohort,(Griffiths et al., 2013) the lack of gender interaction in this analysis reveal that both girls' and boys' participation in VPA are equally affected by socioeconomic and ethnic factors.”

VERSION 2 – REVIEW

REVIEWER	Marco Geraci University of South Carolina
REVIEW RETURNED	11-Mar-2019

GENERAL COMMENTS	The authors have addressed all my previous comments
---